# A Survey of Main Pepper Crop Viruses in Different Cultivation Systems for the Selection of the Most Appropriate Resistance Genes in Sensitive Local Cultivars in Northern Spain

**DOI:** 10.3390/plants11060719

**Published:** 2022-03-08

**Authors:** Mikel Ojinaga, Pedro Guirao, Santiago Larregla

**Affiliations:** 1Plant Production and Protection Department, NEIKER-Basque Institute for Agricultural Research and Development, C/Berreaga 1, 48160 Derio, Spain; mojinaga@neiker.eus; 2Plant Production and Microbiology Department, Universidad Miguel Hernández, 03312 Orihuela, Spain; pedro.guirao@umh.es

**Keywords:** *Capsicum annuum*, multiple infections, *Tobamovirus*, *Tospovirus*, *Potyvirus*, pathotype

## Abstract

Viral diseases have become one of the main phytosanitary problems for pepper growers in the Basque Country (northern Spain). In 2014, a survey was carried out to determine the prevalence of the most common viruses found in Gernika pepper and Ibarra chili pepper landraces. A total of 97 plots were surveyed and classified according to the crop system. Within these plots, 1107 plants were sampled and tested for tobacco mosaic virus (TMV), tomato mosaic virus (ToMV), tobacco mild green mosaic virus (TMGMV), pepper mild mottle virus (PMMoV), paprika mild mottle virus (PaMMV), potato virus Y (PVY) and tomato spotted wilt virus (TSWV) applying a DAS-ELISA test. PaMMV was verified by the non-radioactive molecular hybridization technique and it was found to be negative. All viruses were detected, but the most prevalent viruses were PVY and TMGMV (19.8% and 10.6% of tested plants, respectively). Differences among cultivation systems were found for most of the tested viruses. PVY had a higher level of infection under open field conditions (27.3%) than under greenhouse conditions (12.3%). Inversely, the viruses belonging to the *Tobamovirus* genus and TSWV prevailed under greenhouse conditions (28.9% and 5.2%) when compared to open field (11.2% and 1.1%), respectively. Single (28%) and multiple infections (8.9%) were found. All PMMoV isolates were classified as pathotype P1.2. Survey results indicated that tobamovirus and PVY resistance genes would be the most appropriate to be included in breeding programs with these sensitive pepper landraces.

## 1. Introduction

Pepper is one of the main vegetable crops in the Basque Country (northern Spain), with a cultivated area of 250 ha in 2018. Pepper crops of this region are mainly concentrated in the Atlantic humid temperate climate zones [1]. The two main pepper landraces are Gernika type pepper and Ibarra yellow chili pepper (Appendix A). Roasting pepper landraces are also cultivated but in fewer quantities. Several selected commercial cultivars have been obtained from the different varietal pepper types grown in this area, although all of them are vulnerable to viral infections. The landrace of Gernika is mainly cultivated under greenhouse conditions from March to October. The landrace of chili pepper is mainly cultivated in open field conditions from May to October [2].

Cultivated peppers (*Capsicum* spp.) are relatively susceptible to viruses that can cause important production losses in this crop worldwide [3,4,5,6,7]. At least 68 viruses have been cited in pepper [8], but approximately 20 species, cause damages in pepper crops [9]. These viruses are classified into six genera, namely *Potyvirus, Tobamovirus, Tospovirus, Cucumovirus, Polerovirus* and *Begomovirus* [10]. Moreover, in protected crops in Spain the most frequent viruses are pepper mild mottle virus (PMMoV, *Tobamovirus*), tobacco mosaic virus (TMV, *Tobamovirus*), tomato mosaic virus (ToMV, *Tobamovirus*), potato virus Y (PVY, *Potyvirus*), tomato spotted wilt virus (TSWV, *Toposvirus*), broad bean wilt virus (BBWV, *Fabavirus*), tomato yellow leaf curl virus (TYLCV, *Begomovirus*), tobacco mild green mottle virus (TMGMV, *Tobamovirus*), pepper vein yellows virus (PeVYV, *Polerovirus*) and parietaria mottle virus (PMoV, *Ilarvirus*) [11]. However, in open field conditions, the most important viruses are PVY, cucumber mosaic virus (CMV, *Cucumovirus*) and TSWV. Less frequently, alfalfa mosaic virus (AMV, *Alfamovirus*), BBWV and ToMV have also been found in open field conditions [12]. Furthermore, the official plant health laboratories of the Basque Country have diagnosed several viruses in pepper crops, such as TMV, ToMV, TMGMV, PMMoV, CMV, PVY and TSWV.

The viruses that cause the highest crop losses in the region are the group of tobamoviruses, PVY and TSWV. Firstly, tobamoviruses represent the most important phytosanitary problem for protected pepper crop viability in the study region. These viruses are seed-borne and can be mechanically transmitted very efficiently [4]. *Tobamovirus* resistance in pepper is considered to be provided by four allelic genes (L1, L2, L3 and L4) [13,14]. There are different pathotypes among *Tobamovirus*—P0, P1, P1,2 or P1,2,3-, which are determined by their ability to overcome L gene-mediated resistance [15,16]. Secondly, PVY is widely distributed in open field conditions, but usually causes lower crop yield losses than tobamoviruses in the study area. PVY is transmitted in a non-persistent manner [17,18] by various aphid species [19] and it has three pathotypes, according to their virulence on pepper genotypes [20]. Several PVY resistance genes have been described [21]. Finally, TSWV is also important in protected pepper crops and it is transmitted by *Frankliniella occidentalis* in a persistent propagative manner [22]. A single dominant resistance gene known as Tsw controls the virus [23]. However, the resistance conferred by this gene decreases under high temperature conditions in protected pepper crops [24].

Research on the occurrence, incidence and distribution of viral diseases is essential for developing diagnostic tools and appropriate control measures [3] such as the elimination of viral vectors and the use of resistant cultivars [25]. In our study, pepper crop surveys have focused on those viruses that cause the greatest economic losses and that can also be controlled by resistance genes. The first goal of this study consisted in determining the prevalence and the distribution of the most severe viral diseases infecting pepper crops in the studied region and under different cultivation systems. The second objective was to determine the pathotypes of PMMoV which are present in the region in order to know which resistance genes must be considered for the genetic control of tobamoviruses. This knowledge will be necessary for the selection of the most appropriate resistance genes to include in future breeding programs of these local pepper cultivars.

## 2. Results

### 2.1. Incidence of Virus Infection

A total of 408 of the 1107 individual plant samples (36.9%) tested positive for at least one of the viruses analyzed (Table 1). Under open field, soil greenhouse and hydroponic greenhouse conditions, the percentages of plants with viral infections were 34.1%, 38.3% and 41.1%, respectively. Considering the total percentage of infected samples, there were no significant differences between the type of pepper (χ^2^_2,1107_ = 0.88; *p* = 0.6443) or the type of cultivation system (χ^2^_2,1107_ = 2.55; *p* = 0.2796) (Table 1).

The percentage of plots that showed infection was very high in all the cultivation systems. The average of the plots infected by any of the viruses evaluated was 76.3%. However, no significant differences were observed between the types of peppers (χ^2^_2,97_ = 1.08; *p*= 0.5807) and the cultivation system (χ^2^_2,97_ = 1.09; *p* = 0.5797) (Table 1).

In chili peppers, the percentage of infection was significantly higher (χ^2^_2,593_ = 12.12; *p* = 0.0023) when it was produced under greenhouse conditions (56.1% in soil greenhouses and 62.5% in hydroponic greenhouses) than when it was produced under open field conditions (34.5%) (Table 1).

Significant differences were also found in viral infection among pepper types produced under soil greenhouse (χ^2^_2,290_ = 8.20; *p* = 0.0165) and hydroponic greenhouse (χ^2^_2,263_ = 6.25; *p* = 0.044) conditions. Under the soil greenhouse conditions, the viral infection was 56.1% in chili pepper, 31.5% in Gernika pepper and 45.8% in roasting pepper. In the hydroponic greenhouses, it was 62.5% in chili pepper, 39.1% in Gernika pepper and there was no viral infection in roasting pepper.

### 2.2. Viruses Detected

PVY, TSWV, TMV, ToMV, TMGMV, PaMMV and PMMoV were present in all the cropping systems evaluated (Table 2). Regarding the types of peppers (Table 1), all of the viruses were detected in the chili pepper type and the Gernika type, but TMV and PaMMV were not found in the roasting pepper type (only 51 plants analyzed).

The most frequent viruses were PVY and TMGMV, which were detected in 19.8% and 10.6% of the analyzed plants and in 47.4% and 29.9% of the surveyed plots, respectively (Table 2). The following most relevant viruses were TMV and ToMV with an incidence of 6.2% and 4.7% of the analyzed plants, respectively, and 21.6% of the surveyed plots in both viruses. PMMoV, TSWV and PaMMV showed a lower incidence, with 3.3%, 3.3% and 1.4% of the infected plants and 17.5%, 16.5%, and 10.3% of the plots, respectively. See viruses’ symptoms of surveyed plants in Appendix A.

Of the total number of sampled plants, 753 (68%) were plants with symptoms and 354 (32%) were asymptomatic plants. Among symptomatic plants, 343 (45.6%) were positive for any of the viruses analyzed. In contrast, among asymptomatic plants, 64 (18.1%) were positive. The most important viruses detected among the symptomatic plants were PVY (24.5%) and TMGMV (12.1%), as well as in the asymptomatic plants (8.1 and 8.1%, respectively).

With respect to the cultivation system, among the plants sampled in open field conditions, 509 (91.9%) were symptomatic and 45 (8.1%) were asymptomatic. In this cultivation system, the number of positive plants to viruses was 185 (36.3%) in symptomatic and 2 (4.4%) in asymptomatic plants. In protected crops, 309 (55.9%) were symptomatic plants and 244 (44.1%) were asymptomatic. In this cultivation system, the number of positive plants to viruses was 158 (64.7%) in symptomatic and 62 (20.1%) in asymptomatic plants.

### 2.3. Virus Importance and Cultivation System

There were significant differences in the percentage of infected plants between different cultivation systems for all viruses evaluated, except TMV (χ^2^_2,1107_ = 1.40; *p* = 0.4975) (Table 2). However, the only two viruses that showed significant differences in the percentage of infected plots between different crop systems were PVY (χ^2^_2,1107_ = 7.77; *p* = 0.0205) and TMV (χ^2^_2,1107_ = 7.40; *p* = 0.0247). Plots contaminated by PVY were 77.8% under open field conditions, 40.5% in soil greenhouses and 28.6% in hydroponic greenhouses. Plots infected by TMV were 40.7%, 9.5% and 21.4%, respectively. PVY had a greater presence in the plants taken under open field conditions (27.3%) than in the greenhouse plants (14.8% in soil greenhouses and 9.5% in hydroponic greenhouses) (χ^2^_2,1107_ = 33.28; *p* < 0.0001). Nevertheless, TSWV and all the tobamoviruses, except TMV, showed higher sample positivity in greenhouse conditions (*p* < 0.05) (Table 2).

Viruses belonging to the genus *Tobamovirus* predominated over the rest of the viruses analyzed. These viruses were found in 20.1% of the sampled plants (Table 2), PVY was observed in 19.8% of the surveyed plants and TSWV was present in 3.3% of the plants. Under greenhouse conditions, tobamoviruses had a greater prevalence in the crops (28.9%), while PVY (12.3%) and TSWV (5.2%) showed lower prevalence. However, under open field conditions, the prevalence of PVY (27.3%) was significantly higher than tobamovirus (11.2%) and TSWV (1.1%).

### 2.4. Single and Multiple Virus Infections

Single infections were detected in 28.0% of the analyzed samples, whereas multiple infections were observed in 8.9% of them (Table 3). PVY was the most common virus in single infections, with a 14.2% rate of incidence in the analyzed plants. Single infection with PVY was detected in 22% of the plants in the open field cultivation system, but it was present in only 6.3% of the plants in the greenhouse system. The second most common virus in single infections was TMGMV, which was present in 5.6% of the analyzed plants (Table 3). Single infection of TMGMV was observed in only 0.4% of the analyzed plants under open field conditions, while it was found in 10.8% of the samples under greenhouse conditions. Single infections with PMMoV and PaMMV were the scarcest in this survey, being present in only 1.1% and 0.45% of the analyzed plants, respectively.

A total of 15 combinations of double infection, 10 triple, 4 quadruple and 2 quintuple were observed in the analyzed samples. The two most prevalent viruses (PVY and TMGMV) were observed in 91.4% of double infections, and in 100% of triple, quadruple and quintuple infections (Table 3).

### 2.5. PMMoV Pathotypes Determination Bioassay

All the isolates of PMMoV that were inoculated caused disease in all the plants of the sensitive Negral cultivar, showing typical symptoms of PMMoV (mosaic in upper non-inoculated leaves) (Table 4). These symptoms were confirmed with the positive results obtained by DAS-ELISA at 15 and 30 DAI of PMMoV isolates. However, all the isolates inoculated in the F1 commercial hybrids Padua (with L3 resistance gene to pathotypes P0, P1 and P1.2) and Giulio (with L4 resistance gene to pathotypes P0, P1, P1.2 and P1.2.3) showed symptoms of hypersensitive response similar to NS followed by LA associated with resistance to PMMoV. In the above-mentioned commercial hybrids, all of the inoculated plants at 30 DAI tested negative for the virus by DAS-ELISA. Only two isolates gave positive results by DAS-ELISA in the cultivar Padua but only in one of the five inoculated plants, which were accompanied by hypersensitivity symptoms (NS) in inoculated leaves at 15 DAI. These same two plants gave negative results by DAS-ELISA and showed hypersensitivity symptoms (necrotic spots followed by abscission) in inoculated leaves at 30 DAI (Appendix A).

## 3. Discussion

The main pepper landraces grown in the Basque Country were infected by some of the viruses that are usually the most important in pepper crops, in accordance with other research on pepper crops worldwide [4,8]. In this study, all the viruses analyzed by DAS-ELISA tested positive and viral infections appeared specific to cropping systems [11]. Serological assays generally offer a reliable method in large-scale surveys for the detection of those viruses that are expected to be in high concentrations but may entail the risk of misidentification due to its low specificity [26]. According to our results, it would be the first time that PaMMV has been detected in Spain. However, after verifying it by the non-radioactive molecular hybridization technique, it was found that they were negative (Appendix A). Serological cross reactivity has been observed between some tobamoviruses [27].

In our survey, plants with virus-like symptoms showed negative results for the viruses analyzed by DAS-ELISA. These symptoms could have been caused by the presence of already known pepper viruses not evaluated in this study [25] or even new viruses that have not been discovered yet [28]. Inversely, there have been plants without virus symptoms which tested positive for viruses. This implies a difficulty when establishing control strategies.

Multiple virus infections were found and plants with up to quintuple co-infections were identified. Synergistic viral interactions that result in devastating diseases often occur when two or more unrelated plant viruses infect a plant simultaneously [29]. These multiple infections have additive and multiplicative effects on the severity of the symptoms induced in plants [30,31]. Mixed infections are rather common in nature and occur in areas where there is a diversity of vectors like thrips and aphids, or where a single vector can transmit different virus species [32]. Multiple viral infections have also been identified in *Capsicum* spp. plants using RNA-seq analysis [10,33,34]. The primary inoculums of *Tobamovirus* that may be in soil-substrate organic matter or in infected seed [4] can also contribute to these co-infections. In our study, 68% of the co-infections were found under greenhouse conditions and 70% of these coinfections were associated with TMGMV. Similarly, 93% of the coinfections were associated with PVY under open field conditions.

According to other previous studies, the most frequent virus was PVY, which had a greater presence in open field conditions than in greenhouse conditions [11,32,35,36,37]. The first infections of PVY in the plots are known to be caused by the flights of aphids in most cases [32,38,39]. The primary PVY inoculum can remain in the plots by being hosted on weeds from one season to the next [40]. In the context of our study, the temperate and humid climate of the surveyed area is well suited for weeds, which can become reservoirs for this virus [41,42,43]. Svoboda and Svobodová (2012) showed that winged aphids migrate from weeds to open field pepper crops, potentially contributing to the introduction of non-persistently transmitted viruses in the crops. This highlights the importance of protecting pepper crops from aphids after field transplantation. In line with other studies [11], aphid-borne viruses were rarely important under greenhouse conditions.

Conversely, the incidence of tobamoviruses was greater under greenhouse conditions than under open field conditions. In intensive greenhouse farming systems, the high number of harvests can favor the spread of tobamoviruses in the crops through its mechanical transmission by workers [44], as it occurs with these pepper varieties which are harvested unripened. This is especially important in greenhouses, where the greater density of plants and the best thermal conditions for the virus replication [45] stimulate the viral spread between plants and increase the risk of generating an epidemic.

Tobamoviruses may be spread during transplanting and at harvest after primary infections via infected seeds [32]. The relatively high infection rate of tobamoviruses obtained in the survey is valuable information in order to reduce the risk of tobamoviruses transmission by seed. Therefore, certified virus free seed production systems have been implemented for Gernika pepper and Ibarra chili-pepper cultivars. This process reduces the probability of introducing primary inoculum with seeds. Thus, seed disinfestation with thermotherapy or with chemical products should be one of the measures to produce virus-free plants [46,47]. However, some farmers still use self-produced non-disinfected seeds which could cause crop epidemics [36].

In this survey, the TMGMV incidence was higher in hydroponic greenhouses than in soil greenhouses. These results appear to be well substantiated by a higher transmission of tobamoviruses through the nutrient solution in the hydroponic cultivation system [48]. It was shown that ToMV can be released from plant roots, survive in nutrient solution, and infect other plants (causing symptoms) through roots without root contact [49].

The use of cultivar resistance against tobamoviruses might be an effective method to control these viral diseases [50] in the aforementioned pepper landraces. In the pathotype determination bioassay, all the analyzed PMMoV samples belonged to pathotype P1.2. Therefore, it would be sufficient to introduce the L3 gene in the landraces cultivated in this area to achieve resistance to all the tobamovirus strains that have been sampled [51,52]. In two of the thirty-six PMMoV isolates that were inoculated in the cultivar with resistance gene L3 (cultivar Padua), a transient systemic infection occurred in only one of the five inoculated plants. However, it was accompanied by virus resistance HR symptoms. The transient systemic infection was confirmed by the ELISA positive result of the first analysis and by the negative result of the second analysis. These facts would indicate an initial virus multiplication in the inoculated plants that was ultimately controlled by the L3 resistance gene. The early phenological state of inoculation and the high load of the inoculated virus can overcome the resistance to the virus [52]. The plant activates mechanisms to control the virus, but since the plant is small and the viral load is high, the virus can translocate to tissues far from the point of inoculation, causing a systemic hypersensitive response (SHR) [53,54,55,56]. This mechanism may be due to a late biochemical and physiological response associated with HR [54].

As occurred with the tobamoviruses, TSWV incidence was greater under protected crops. The highest incidence of TSWV in this cultivation system could be explained by the lack of insect screens in the greenhouses, the transmission mode of this virus, and the faster life cycle of the insect vector [22]. The mild climate conditions of the Basque Country favor the development of thrips and the vectored TSWV in greenhouses. Our results are in good agreement with previous studies which reported high incidence of TSWV in protected pepper crops located in mild climate regions [11].

Regarding the cultivation system and virus control strategies, under soil greenhouses, the solution to inactivate tobamoviruses would be to perform biodisinfestation with organic amendments or to compost crop residues for the control of other soil pathogens [57,58,59]. Under hydroponic greenhouses, the tobamoviruses load could be lowered by exchanging the substrate bags, removing crop residues and disinfecting the irrigation system (heat treatments, ultra-violet radiation and ozonisation) [60,61]. Finally, in the case of viruses vectored by aphids under open field cultivation, the control of vector insects, virus reservoir adventitious plants and an appropriate planting date could contribute greatly to reducing the risk of epidemics [32,36,62].

The current status of virus infection in the surveyed area, and the use of landraces or cultivars without virus resistance genes, could promote virus outbreaks in pepper crops. Therefore, backcross breeding programs have been started in order to introduce virus resistance genes to tobamoviruses [63], PVY and TSWV in Gernika pepper and Ibarra chili pepper. To ensure that the genetic resistance introduced is durable and not overcome by viruses, it is important to combine cultivar resistance with other control measures such as hygiene measures to lower the viral load and produce virus-free certified seed.

## 4. Material and Methods

### 4.1. Crop Survey

In October 2014, at the end of the crop cycle, a survey of viruses was carried out in 44 counties of the Basque Country (Figure 1). In this survey, 77% of the professional pepper crop area of the region was prospected. The plant samples were collected individually and put in closed polyethylene bags in refrigerated containers. They were brought to the laboratory the same day, where they were kept at 5 °C until analyzed. The number of samples per plot was determined by the area of the pepper plots. In plots smaller than 1000 m^2^ (61.8%), 6 samples were surveyed; 10 samples were sampled in plots between 1000–2000 m^2^ (18.8%); 16 samples were surveyed in plots between 2000–4000 m^2^ (12.3%) and 40 samples in plots larger than 4000 m^2^ (7.1%). Each crop plot was completely surveyed and leaves showing virus-like symptoms were given preference when collecting samples with the aim of knowing which viruses are the ones that create the greatest damage in pepper crops in the study area. In those plots with not enough samples with symptoms, asymptomatic plants were randomly sampled.

A total of 97 plots were surveyed, which represented an area of 24.3 ha. The surveyed surface corresponded to the open field cultivation system (78%), soil greenhouses (7%), and hydroponic greenhouses (15%). Out of 1107 plants analyzed, 50% were taken under open field conditions and 50% under greenhouse conditions. Under protected conditions, 53% of the samples were taken in soil greenhouses and 47% of the samples were taken in hydroponic greenhouses. Out of the 1107 samples, 53% were samples of chili pepper type, 42% were Gernika pepper type and 5% were roasting pepper type (Table 1).

### 4.2. Sample Analysis

The samples were tested by the double antibody sandwich-enzyme-linked immunosorbent assay (DAS-ELISA) [64] technique using specific antisera against TMV, ToMV, TMGMV, PaMMV, PMMoV, PVY and TSWV. Antibodies for all viruses were commercially provided by Loewe Biochemica (DE). The absorbance of the serological reaction was measured using a Model 550 Microplate Reader (Bio Rad, Hercules, CA, USA). Samples were considered to be infected when absorbance was three times greater than the average absorbance of the negative controls from healthy peppers.

The significance of differences between the percentages of infected plants and infected plots among pepper type and crop system was tested using Chi-square (χ^2^) tests for equality of distributions. The statistical significance level was fixed at *p* < 0.05.

### 4.3. PMMoV Pathotype Determination Bioassay

PMMoV pathotypes resistance genes confer resistance to the rest of viruses belonging to the group of tobamoviruses [13] which are present in the study area. Therefore, we performed a bioassay for the determination of the pathotypes of PMMoV present in the Basque Country.

The leaf samples in which PMMoV infection was detected (36 samples) were frozen at −80 °C. These samples were inoculated in three different pepper cultivars: “Negral” (susceptible to all pathotypes of *Tobamovirus*), “Padua RZ F1” (with L3 gen; resistant to pathotypes P0, P1 and P1.2) and “Giulio RZ F1” (with L4 gen; resistant to pathotypes P0, P1, P1.2 and P1.2.3). Furthermore, in each of the three pepper cultivars, three control treatments were included in the inoculation test: Ctr-: Negative control of non-inoculated plants; Ctr + P1.2: Positive control of an isolate belonging to PMMoV pathotype P1.2; Ctr + P1.2.3: Positive control of an isolate belonging to PMMoV pathotype P1.2.3.

Each of the PMMoV positive samples was inoculated in 5 plants of each cultivar at the stage of 6 fully developed true leaves. Virus inoculum suspension was prepared with 10 mL of phosphate buffer (NaH2PO4—Na2HPO4) 50 mM at pH 7.5 and 1 g of frozen fresh leaf tissue which had been previously crushed in a mortar to achieve homogenization. Carborundum powder was added for an abrasive effect. Tested plants were inoculated by rubbing the two true basal leaves with the virus inoculum suspension. For the negative control of non-inoculated plants, a suspension was prepared with the tissue of healthy pepper leaves, phosphate buffer, and carborundum. After inoculation, plants were kept in a growth chamber maintained at 25 °C during the day and 20 °C at night, with a 14 h photoperiod. Symptoms were recorded in each of the inoculated plants at 15 and 30 days after inoculation (DAI). When plants were sensitive to the virus, mosaic (M) symptoms were observed on the non-inoculated upper leaves. The plants were considered resistant when necrotic spots (NS), followed by leaf abscission (LA), were observed in the inoculated leaves [23]. These symptoms were considered to be the result of the plant hypersensitive response (HR) at the point of *Tobamovirus* penetration. The HR is a generalized response in plants against pathogens, characterized by the isolation of the virus through local necrotic lesions at the point of infection, thus preventing infection of the plant [65]. The DAS-ELISA test was used to determine the presence or absence of PMMoV at 15 and 30 DAI.

## 5. Conclusions

The tobamovirus group was predominant in protected crops of the Basque Country, while PVY was predominant in open field conditions. Within the tobamoviruses, TMGMV was the most important. TSWV was also more important in protected cultivation. The seven analyzed viruses (PVY, TSWV, TMV, ToMV, TMGMV, PaMMV and PMMoV) tested positive by DAS-ELISA in all the evaluated cropping systems. However, when PaMMV was analyzed by the non-radioactive molecular hybridization technique, it was found to be negative. It may be possible that it was cross-positive with other tobamoviruses. Various combinations of multiple infections were found, with coinfections of up to five viruses.

All PMMoV isolates studied in the bioassay belong to the pathotype P1.2 which could be controlled with the L3 resistance gene in the sensitive pepper landraces grown in the region.

The application of anti-insect mesh could be effective to reduce the risk of infection by PVY or TSWV in open field and greenhouse cultivation systems, respectively. Crop infection by tobamoviruses in the greenhouse cultivation system could be reduced by the use of resistance genes combined with the use of certified virus-free disinfested seed and the inactivation of virus inoculum in the crop residues.

## Figures and Tables

**Figure 1 plants-11-00719-f001:**
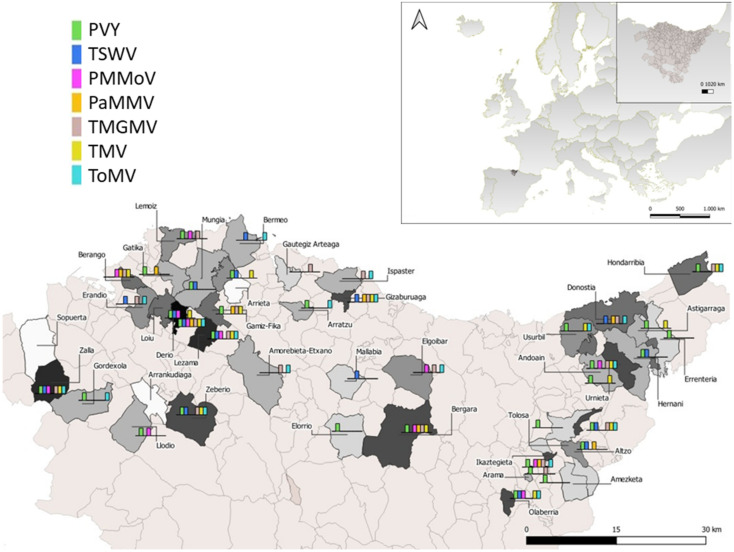
Map of the Basque Country (northern Spain) showing the areas and the distribution of viruses on pepper (*Capsicum annuum* L.) plants in each surveyed county in 2014. PaMMV was detected by DAS-ELISA, but after being analyzed by the non-radioactive molecular hybridization technique, it was found to be negative (Appendix A).

**Table 1 plants-11-00719-t001:** Number and percentage of individually sampled plants (above) and surveyed plots (below) that tested positive for at least one of the viruses analyzed by DAS-ELISA, depending on the pepper type, the cultivation system and probability of the χ^2^ test for equality of proportions among the different pepper types and crop systems.

	Open Field		Soil Greenhouse		Hydroponic Greenhouse		Total		χ^2^ Value (Cultivation System)	*p* Value (Cultivation System)
Pepper Type	Plants	% Plants Infected	Plants	% Plants Infected	Plants	% Plants Infected	Plants	% Plants Infected		
Chili pepper	495	34.5	66	56.1	32	62.5	593	38.4	12.12	0.0023
Gernika pepper	38	28.9	200	31.5	225	39.1	463	35.0	2.18	0.3355
Roasting pepper	21	33.3	24	45.8	6	0.0	51	35.3	2.89	0.2351
Total	554	34.1	290	38.3	263	41.1	1107	36.9	2.55	0.2796
χ^2^ value (Pepper type)	0.33		8.20		6.25		0.88			
*p* value (Pepper type)	0.8486		0.0165		0.044		0.6443			
	**Open Field**		**Soil Greenhouse**		**Hydroponic Greenhouse**		**Total**		**χ^2^ Value (Cultivation System)**	** *p* ** **Value (Cultivation System)**
**Pepper Type**	**Plots**	**% Plots Infected**	**Plots**	**% Plots Infected**	**Plots**	**% Plots Infected**	**Plots**	**% Plots Infected**		
Chili pepper	21	90.5	10	90.0	4	75.0	35	88.6	0.09	0.9541
Gernika pepper	4	100.0	28	67.9	23	65.2	55	69.1	0.61	0.7374
Roasting pepper	2	50.0	4	100.0	1	0.0	7	71.4	1.30	0.5219
Total	27	88.9	42	76.2	28	64.3	97	76.3	1.09	0.5797
χ^2^ value (Pepper type)	0.40		0.80		0.72		1.08			
*p* value (Pepper type)	0.8180		0.6693		0.6983		0.5807			

**Table 2 plants-11-00719-t002:** Number and percentage of sampled plants (above) and surveyed plots (below) that tested positive for each of the viruses analyzed by DAS-ELISA in each crop system, and probability of the χ^2^ tests for equality of proportions among the different crop systems.

	Open Field	Soil Greenhouse	Hydroponic Greenhouse	Total	χ^2^ Value	*p* Value
Number of plants	554	290	263	1107		
Virus infection (%)						
PVY	151 (27.3%)	43 (14.8%)	25 (9.5%)	219 (19.8%)	33.28	<0.0001
TSWV	6 (1.1%)	14 (4.8%)	16 (6.1%)	36 (3.3%)	16.71	0.0002
TMV	36 (6.5%)	14 (4.8%)	19 (7.2%)	69 (6.2%)	1.40	0.4975
ToMV	10 (1.8%)	32 (11%)	10 (3.8%)	52 (4.7%)	35.11	<0.0001
TMGMV	10 (1.8%)	44 (15.2%)	63 (24.0%)	117 (10.6%)	90.66	<0.0001
PaMMV ^a^	2 (0.4%)	6 (2.1%)	7 (2.7%)	15 (1.4%)	8.44	0.0147
PMMoV	13 (2.3%)	9 (3.1%)	15 (5.7%)	37 (3.3%)	6.08	0.0478
	**Open Field**	**Soil Greenhouse**	**Hydroponic Greenhouse**	**Total**	**χ^2^ Value**	***p* Value**
Number of plots	27	42	28	97		
Virus infection (%)						
PVY	21 (77.8%)	17 (40.5%)	8 (28.6%)	46 (47.4%)	7.77	0.0205
TSWV	5 (18.5%)	5 (11.9%)	6 (21.4%)	16 (16.5%)	1.02	0.6014
TMV	11 (40.7%)	4 (9.5%)	6 (21.4%)	21 (21.6%)	7.40	0.0247
ToMV	4 (14.8%)	12 (28.6%)	5 (17.9%)	21 (21.6%)	1.70	0.4278
TMGMV	4 (14.8%)	13 (31%)	12 (42.9%)	29 (29.9%)	3.64	0.1617
PaMMV ^a^	2 (7.4%)	4 (9.5%)	4 (14.3%)	10 (10.3%)	0.68	0.7135
PMMoV	5 (18.5%)	7 (16.7%)	5 (17.9%)	17 (17.5%)	0.03	0.9828

^a^ PaMMV was detected by DAS-ELISA, but after being analyzed by the non-radioactive molecular hybridization technique, it was found to be negative (Appendix A).

**Table 3 plants-11-00719-t003:** Single or mixed viral infections in each pepper crop system expressed as percentage of plants that were positive for each of the viruses analyzed by DAS-ELISA.

	Infections Detected				
Viruses Detected	Openfield (%), n = 554	Soil Greenhouse (%), n = 290	Hydroponic Greenhouse (%), n = 263	Total Greenhouse (%), n = 553	Total (%), n = 1107
Single infections					
PVY	22.0	7.6	4.9	6.3	14.18
TSWV	0.5	4.5	3.8	4.2	2.35
TMV	3.2	0.3	1.9	1.1	2.17
ToMV	1.4	5.2	0.4	2.9	2.17
TMGMV	0.4	6.9	15.2	10.8	5.60
PaMMV ^a^	0.0	0.7	1.1	0.9	0.45
PMMoV	0.9	1.4	1.1	1.3	1.08
Total	28.5	26.6	28.5	27.5	28.00
Double infections					
PVY + TSWV	0.4	0.0	0.4	0.2	0.27
PVY + TMV	2.7	0.7	0.0	0.4	1.54
PVY + ToMV	0.2	0.7	0.0	0.4	0.27
PVY + TMGMV	0.5	0.7	2.3	1.4	0.99
PVY + PaMMV ^a^	0.2	0.3	0.0	0.2	0.18
PVY + PMMoV	0.5	0.7	0.0	0.4	0.45
TSWV + TMV	0.0	0.0	0.4	0.2	0.09
TSWV + ToMV	0.0	0.0	0.4	0.2	0.09
TSWV + PaMMV	0.0	0.0	0.4	0.2	0.09
TMV + ToMV	0.0	0.0	0.8	0.4	0.18
TMV + TMGMV	0.0	0.7	1.1	0.9	0.45
TMV + PMMoV	0.0	0.0	0.4	0.2	0.09
ToMV + TMGMV	0.0	2.1	0.4	1.3	0.63
TMGMV + PaMMV ^a^	0.0	1.0	0.0	0.5	0.27
TMGMV + PMMoV	0.2	0.3	2.3	1.3	0.72
Total	4.7	7.2	8.7	8.0	6.32
Triple infections					
PVY + TSWV + PMMoV	0.2	0.0	0.0	0.0	0.09
PVY + TMV + ToMV	0.2	0.3	0.4	0.4	0.27
PVY + TMV + TMGMV	0.0	1.0	0.4	0.7	0.36
PVY + TMV + PMMoV	0.0	0.3	0.4	0.4	0.18
PVY + ToMV + TMGMV	0.0	1.0	0.0	0.5	0.27
PVY + TMGMV + PMMoV	0.2	0.0	0.0	0.0	0.09
TMV + ToMV + TMGMV	0.0	0.0	0.8	0.4	0.18
TMV + TMGMV + PMMoV	0.2	0.0	0.0	0.0	0.09
ToMV + TMGMV + PMMoV	0.0	0.3	0.0	0.2	0.09
TMGMV + PaMMV ^a^ + PMMoV	0.0	0.0	0.8	0.4	0.18
Total	0.7	3.1	2.7	2.9	1.81
Quadruple infections					
PVY + TSWV + TMV + ToMV	0.0	0.3	0.0	0.2	0.09
PVY + TMV + ToMV + TMGMV	0.0	1.0	0.0	0.5	0.27
PVY + TMGMV + PaMMV ^a^ + PMMoV	0.0	0.0	0.4	0.2	0.09
TSWV + TMV + ToMV + TMGMV	0.0	0.0	0.4	0.2	0.09
Total	0.0	1.4	0.8	1.1	0.54
Quintuple infections					
PVY + TSWV + TMV + ToMV + PMMoV	0.0	0.0	0.4	0.2	0.09
PVY + TMV + TMGMV + PaMMV ^a^ + PMMoV	0.2	0.0	0.0	0.0	0.09
Total	0.2	0.0	0.4	0.2	0.18

^a^ PaMMV was detected by DAS-ELISA, but after being analyzed by the non-radioactive molecular hybridization technique, it was found to be negative (Appendix A).

**Table 4 plants-11-00719-t004:** PMMoV isolates pathotype determination in the inoculation bioassay with a set of plants of different pepper genotypes.

(x/5) ^a^	15 Days			30 Days		
	N ^c^	P ^c^	G ^c^	N	P	G
0/5	Ctr-(_)	Ctr-(_)	Ctr-(_)	Ctr-(_)	Ctr-(_)	Ctr-(_)
		Ctr + P_1_._2_ (NS)	Ctr + P_1_._2_ (NS)		Ctr + P_1_._2_ (NS; LA)	Ctr + P_1_._2_ (NS; LA)
		Ctr + P_1_._2_._3_ (NS)_3_	Ctr + P_1_._2_._3_ (NS)			Ctr + P_1_._2_._3_ (NS; LA)
		34 (NS ^d^)	36 (NS)		36 (NS; LA ^d^)	36 (NS; LA)
1/5		2 (NS)				
2/5		Ctr + P_1_._2_._3_ (NS)_2_				
3/5	1 ^b^ (M ^d^)					
4/5	4 (M)					
5/5	Ctr + P_1_._2_ (M)			Ctr + P_1_._2_ (M)		
	Ctr + P_1_._2_._3_ (M)			Ctr + P_1_._2_._3_ (M)	Ctr + P_1_._2_._3_ (M)	
	31 (M)			36 (M)		

Each of the 36 PMMoV isolates from positive samples collected in the survey was inoculated in five plants of each cultivar in the six fully-developed leaf stage. Three control treatments were included in the inoculation test: Ctr-: Negative control of non-inoculated plants; Ctr + P_1.2_: Positive control of an isolate belonging to PMMoV pathotype 1.2; Ctr + P_1.2.3_: Positive control of an isolate belonging to PMMoV pathotype 1.2.3. ^a^ x is the number of plants with positive results with DAS-ELISA at 15 and 30 days after inoculation with the 36 isolates of PMMoV collected in the survey. ^b^ The number of isolates of PMMoV which showed the same pattern of DAS-ELISA results in the inoculated plants. ^c^ Cultivars: N= Negral, sensitive landrace to all the pathotypes of *Tobamovirus*; P = Padua RZ F1 commercial hybrid with L3 resistance gen to *Tobamovirus* pathotypes P_0_, P_1_ and P_1.2_; G= Giulio, RZ F1, with L4 resistance gen to pathotypes P_0_, P_1_, P_1.2_ and P_1.2.3_. ^d^ Symptoms: NS: Necrotic spots in inoculated leaves; LA: leave abscission in inoculated leaves; M: Mosaic in upper non-inoculated leaves; (_): No symptoms. The symptoms obtained in the inoculation test are shown in parentheses. The subscript numbers outside the parentheses indicate the number of plants in the Ctr + P_1.2.3_ treatment that showed the same type of symptoms but with different results with DAS-ELISA. See symptoms of inoculated plants in Appendix A.

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
