# Peer review of "A Survey of Main Pepper Crop Viruses in Different Cultivation Systems for the Selection of the Most Appropriate Resistance Genes in Sensitive Local Cultivars in Northern Spain"

_plants, 2022, doi:10.3390/plants11060719_

Round 1
Reviewer 1 Report
Your paper looks good, however, the virus identification was based on serological tests (DAS-ELISA) using polyclonal antibodies, and 4 characterized/analyzed viruses are belong to the same genus (Tobamovirus), accordingly, molecular characterization for few isolates of these viruses is needed to improve the results (especially the samples/isolates with Double, Triple, Quadruple and Quintuple infections; multiple infections is 8.9%).
Please keep attention for the references and follow the journal style.
Author Response
Dear reviewer 1,
In the attached document we have added the answers to your coments.
All the best,
Santiago Larregla

Reviewer 2 Report
The manuscript entitled “A Survey of Main Pepper Crop Viruses for the Selection of the Most Appropriate Resistance Genes to Be Included in Sensitive Local Cultivars in Northern Spain” by Ojinaga and colleagues presents results from a large-scale survey of pepper viruses infecting local varieties in Basque. Moreover, they conducted pathotype determination assays using PMMoV isolates and they compared the incidence of the tested viruses in various cropping systems and in three local pepper cultivars. Unfortunately, I found the manuscript lacking of flow and connectivity, mainly in the introduction and the discussion part, that might had allowed the reader to follow the thread of the obtained information from one sentence to the next and from one paragraph to another. In addition, there was a mass of information that could be omitted, and, on the other hand, a few key points were missing from the manuscript. For these reasons, I recommend rejection of this manuscript with encouragement for resubmission. I also made comments (see below), which I believe they will help the authors highlight their work. Overall, the authors performed an important survey and recorded the status of major viruses in pepper crops, and these information need to be published in the near future. I strongly believe that a revised version will significantly improve the quality of the manuscript and reward the authors for all this work.
Major comments
Title: since you did not evaluate pepper cultivars with resistance genes on all these viruses in order to be included in the cropping systems of Basque, I would recommend to change this title and keep it focused on the survey and incidence of pepper viruses in open-field and green house crops, or something like that. I find the introduction part too extensive and a bit chaotic in structure and information. For example L55-69 I believe that the introduction part should had included information on the pepper crop in Basque Country (paragraph one), a list of important viruses that infect pepper crops (worldwide, or specified to Europe, or even more specified to Spain) (paragraph two) and another paragraph on the cropping systems in these landrace. Last, one paragraph is needed for the purpose of this study and for what has been done. Instead, there was a mass of information that was not structured properly and in an understandable way. L144-148: If I understand correctly, both symptomatic and asymptomatic samples were used? Do you have data on the percentage of symptomatic and asymptomatic plants depending on the region? For example, in the intro you mention that “the most prevalent viruses were PVY and TMGMV (19.8% and 10.6% of tested plants)”. How many of these samples exhibited symptoms of viral infections? It is important to know how these percentages are associated with symptoms or not. And did you find viruses in asymptomatic plants? Try to address this gap in the manuscript if possible.
Table 2. You show the number of plots/plants in numbers and then you present in percentages, thus it seems a bit confusing. I would prefer to see the percentages in parentheses and to see the real number of virus infected plants (or plots) in this table. It is easier for the reader to see that from the 551 open field samples, the 200 (…%) are positive (an hypothetical example)
Discussion L348-352: you have a separate paragraph only 5 lines highlighting the contribution of HTS analysis on virus identification in pepper plants. I find this information irrelevant to the concept of you manuscript, since you did not deal with that, or you did not incorporate that, for example, in the last paragraph of the discussion as a promising technology for future diagnostics, thus I think you should delete that. L353-357: I do not believe it is correct to say that you had viral symptoms on plants that that these plants may be a result of abiotic stress. You may point out that you had virus-like symptoms in a few pepper plants such as….., that may be confused with viral infections. For example, interveinal yellowing of leaves may be caused by criniviruses but also by magnesium deficiency. Do you have any examples to support that hypothesis? I do not know any abiotic stess factors that induce chlorotic/necrotic ringspots, or mosaic and mottling (for example). My personal belief (based on extensive research work) is that you may have dealt with other already known pepper viruses (or viroids/phytoplasma) that you did not check in this present work, or even new viruses that haven’t discovered yet.
Finally, what was the percentage of symptomatic plants that were negative to these viruses and, inversely, what was the percentage of asymptomatic plants that were found positive to these viruses? L375. These two paragraphs recommend control measures that need to be taken into account. I find them again too extensive since you did not study the control of these viruses. You did not discuss your results in these paragraphs. As such, you may shorten these paragraphs into 2-3 phrases that may also be transferred to the last paragraph of discussion (for example). Overall questions -Why did you selected PMMoV in the pathotype determination assay and not other tobamoviruses? -What is the status of pepper viruses in Spain? How your work enriched the information on pepper virus distribution in Spain? You mention that this work also shows the first report of PaMMV in pepper fields in northern Spain. Just Northern Spain or Spain? What about the other viruses? Were they present in Spain? Or did anyone conducted a survey of these viruses in Basque? These are questions that I did not see them answered adequately in the manuscript.
Minor comments
L34. Viruses are…
L42. At least 68 viruses (or virus species) have… 43. What do you mean by taxonomic groups? 47. Pepper, along with…, are one …
L70-72 may be omitted
L139. Replace “inside of” with “in” L159. Add “Table 1” at the end of this sentence.
L166: was three times greater than… L168-171: It is not clear to me which two percentages you compare. Also, when you say “The χ2 test was performed to compare the difference between the proportions”, which proportions do you mean? Is it a separate x2 test from the previous one you mention in L168-170?
L222. Delete “All of the analyzed viruses were detected” as it is a repeat in the next phrase. L235. PMMoV, TSWV and PaMMV showed…
L245-246. PVY had a… L250-251. “the highest positivity” it does not sound correct.
L253. “tobamovirus genera was the virus group” also the same. You may say “tobamoviruses had…”
L256. tobamoviruses, not Tobamoviruses
L250 viruses belonging to the genus Tobamovirus L259. PVT was the most common virus.. L289. Delete “see symptoms of …”, just add “Figure S2” at the end of the appropriate sentences L295. Were infected by several viruses
Author Response
Dear reviewer 2,
In the attached document we have added the answers to your coments.
All the best,
Santiago Larregla

Reviewer 3 Report
The present manuscript presented a detailed survey about the incidence of different viruses damaging pepper crops in a certain pepper growing area. The research work presented in the manuscript is fair enough to be published in the journal plants. However, the authors should address some of the comments given below.
Comments:
Why did the authors select those viruses? There are some important viruses are missing in the survey such as the Yellow leaf curl virus Cucumber mosaic virus etc. I am pretty sure the viruses I mentioned must be prevailing in the same growing area. The authors also mentioned them in Line 70-82.
Please check the italicization of the scientific names throughout the manuscript. One example is Line 26 where Capsicum annuum should be italic.
The reference style is not according to the MDPI journal, please change them.
It would be of great interest to the readers to see the pictures of Gernika type pepper and Ibarra yellow chili pepper as well as virus-infected plants. So please add some of the selected pictures even in supplementary data.
Is there any reason that authors preferred the ELISA test for 7 viruses in all 1107 samples when the PCR assays are also available and more convenient to detect the viruses?
Please add the conclusions section with clear results of the survey (Take home message) and solid suggestions to incorporate the resistance in the regional materials. The conclusion should also suggest the best cultivation system to avoid virus infection.
Author Response
Dear reviewer 3,
In the attached document we have added the answers to your coments.
All the best,
Santiago Larregla

Round 2
Reviewer 2 Report
The authors made an effort to improve the structure of the manuscript and reshaped it, following the reviewers' suggestions. I carefully read the manuscript and I believe it has been revised to take account of the raised issues. Nevertheless, there are still a few issues that need to be taken solved before final publication. Therefore, my recommendation for this paper is acceptance with revision.
Major comments
1. L387-388. What do you mean “this information needs to be contrasted with PCR analysis”? Do you mean that you need to verify the presence of PaMMV with PCR? You also state that in L537-539, and you mention the possibility of cross-reaction with DAS-ELISA. As you may imagine, this is a major issue. You present all these results on PaMMV occurrence in single and mixed infections but if there is such possibility, you have at least 2 options: 1. Verify the presence PaMMV with RT-PCR (a few indicative samples), using specific primers for the virus so as to avoid cross-reaction with closely related viruses. 2. Delete all the data on PaMMV (which is not desirable).
2. Structure of discussion
I found the structure of discussion improved and the authors did consider the reviewers’ comments, but, yet, the result is not quite satisfying. For example, you comment on PaMMV (tobamovirus) on 386-388 (1st paragraph), then you mention “the genus tobamoviruses” on L404 (3rdparagraph, first sentence) but you only analyze TSWV, then you comment on tobamoviruses and the intensive greenhouse farming system on L414-420 (4th paragraph), then you mention them again on L491-504 (9th paragraph). Why didn’t you discuss the results on tobamoviruses in 2-3 consecutive paragraphs? As you may see, there is still lack of flow on the most important part of the MS. I still believe it is worth investing one more time in re-shaping the discussion, keeping these things in mind.
3. Conclusions (optional)
In this part, you do not repeat the results (one example L533-536), but you draw conclusions from your results, and this section in optional in cases the discussion is unusually long or complex.
Minor comments
L524. “the tobamovirus group” or “tobamoviruses”
L525. Delete “virus:
L404-405. Rephrase “The genus Tobamovirus was greater”. Do you mean tobamoviruses? Or member in the genus?
L140. Delete “that are grown”
L143. Replace “..are sensitive to viruses”. Maybe ..”are vulnerable/prone to viral infections.”
L144. “..is professionally cultivated primarily…” rephrase
L150. “these species are classified/grouped into 6 genera, namely Potyvirus, …., Begomovirus.”
L151-157. The first letter of each virus is not capital. You may re-visit virus nomenclature for these issues.
L158. “.. the most important viruses are..”
L258-259. “these symptoms were considered to be the result of the plant hypersensitive response (HR) at the point of tobamovirus penetration”
L382-383. Do references [4,8] correspond to pepper landraces in Basque Country? If yes omit my comment.
I just observed that the research manuscript sections on Plants (MPDI) are Introduction, Results, Discussion, Mats& Meths, so maybe you should reorganize it according to the format of PLANTS.
Author Response
See uploaded documents
